# Factors Affecting the Spread, Diagnosis, and Control of African Swine Fever in the Philippines

**DOI:** 10.3390/pathogens12081068

**Published:** 2023-08-21

**Authors:** Chia-Hui Hsu, Rachel Schambow, Maximino Montenegro, Ruth Miclat-Sonaco, Andres Perez

**Affiliations:** 1Center for Animal Health and Food Safety, College of Veterinary Medicine, University of Minnesota, Saint Paul, MN 55108, USA; scham083@umn.edu (R.S.); aperez@umn.edu (A.P.); 2Pig Improvement Company (PIC) Philippines, Pasig City 1605, Philippines; max.montenegro@genusplc.com; 3National Livestock Program, Office of the Undersecretary for Livestock, Department of Agriculture, Elliptical Road, Diliman, Quezon City 1101, Philippines; rmiclat@ati.da.gov.ph

**Keywords:** African Swine Fever, disease control, conjoint analysis, SWOT analysis, risk factors, the Philippines

## Abstract

African Swine Fever (ASF) is a highly contagious disease that threatens the swine industry globally. Since its introduction into the Philippines in 2019, ASF has spread extensively in both commercial and backyard farms. Here, using a mix of qualitative and quantitative methods, including conjoint and SWOT analyses, world café discussions, and multivariable regression models, the most important factors that influence the spread, diagnosis, and control of ASF in the Philippines were identified. Research findings suggest that swill or contaminated feed, inadequate biosecurity protocols, and movement of personnel were the top risk factors favoring ASF spread among farms in general. For commercial farms, contaminated vehicles and personnel were also important, whereas for backyard farms, the introduction of new pigs, environmental contamination, and poor feeding quality were relevant risk factors. Notable clinical signs of ASF in pigs include reduced feed intake, huddled behavior, and reluctance to stand. This study highlights the need for timely reporting, trust-building initiatives, and enhanced biosecurity measures to effectively manage ASF outbreaks in the country. Results here contribute to the knowledge of factors affecting ASF spread in the Philippines and can help design prevention and control measures in ASF-infected countries while enhancing preparedness in countries free from the disease.

## 1. Introduction

African Swine Fever (ASF) is a hemorrhagic disease of swine caused by infection with the ASF virus (ASFV), a double-stranded DNA arbovirus [1,2]. ASF is a disease reportable to the World Organisation for Animal Health (WOAH) due to the significant risk it poses to the global swine industry. ASF only infects swine and can cause up to 100% mortality. Pigs infected by ASF may exhibit clinical signs such as anorexia, high fever, hemorrhages in the skin, bloody diarrhea, pneumonia, abortion in pregnant sows, and sudden death [1,2]. Since its detection outside of Africa in 2007, ASF p72 genotype II has spread rapidly across Europe and Asia, notably being identified in China in 2018 [3]. Subsequently, from 2019 to 2023, it has spread throughout many Southeast Asian countries including Vietnam (2019), Cambodia (2019), Laos (2019), the Philippines (2019), Myanmar (2019), Timor-Leste (2019), Indonesia (2019), Malaysia (2021), Thailand (2022), and Singapore (2023).

Since its first appearance in the province of Rizal [4], the Philippines in July 2019, ASF has swiftly propagated across the country, causing extensive outbreaks in all 17 administrative regions. Spread followed a seasonal pattern, with ASF outbreaks being more frequent in the latter half of each year [5]. Pork production in the Philippines was heavily impacted. In 2021, there was a 41.7% reduction (approximately 2.01 million heads) of the total registered population on commercial pig farms, with subsequent inflation of pork prices [6]. Backyard farms suffered losses as well, dropping from 7.97 million head in 2020 to 6.91 million in the third quarter of 2021. In March 2023, the country’s registered swine population reached approximately 10.18 million heads, roughly 20.5% lower compared to the same quarter in 2020 [6,7]. This ongoing situation, transitioning from epidemic to endemic status for the disease, has raised significant concerns among the general public. These concerns include increasing pork prices and, for some consumers, food security, despite ASFV not infecting humans [8]. To effectively address the ASF epidemic and ensure continued commercial activity in the swine market, there is a need to identify the distinctive risk factors associated with the Philippine context and allocate resources to selectively target the most impactful factors and risks.

As in many countries affected by ASF, much of the data and experience in controlling the disease has not been formally recorded. Qualitative tools and analyses can support the elicitation of the opinion of individuals that have been engaged in control activities to extract knowledge and evidence on the risk factors affecting ASF in the Philippines. Conjoint analysis is a research method that offers a useful approach to understanding the preferences and trade-offs made by stakeholders [9,10]. Originally, this tool was developed for market research to understand consumer preferences, but it has been adopted for use in veterinary sciences to quantify the preferences of respondents and conduct risk assessments [11,12]. Conjoint analysis is particularly useful in that it can help estimate the relative importance of various factors when a large number of variable combinations are present. Rather than evaluate every attribute combination (which would be virtually impossible with high numbers of attributes and levels), statistical methods are used to select a representative subset of attribute combinations. These combinations are used to produce a survey where participants commonly rank or rate the options presented. Survey responses are typically analyzed using regression techniques, which vary depending on the survey method used [10]. In the case of ASF in the Philippines, because data on ASF spread is often lacking or unavailable, conjoint analysis can aid the quantitative estimation of the relative importance of risk factors in ASF transmission using expert stakeholder opinion.

Strengths, weaknesses, opportunities, and threats (SWOT) analysis can also support the identification of risk factors. SWOT analysis is a qualitative technique commonly used to systematically evaluate the advantages and disadvantages of a program [13]. Strengths and weaknesses represent internal attributes of the program itself that may affect its success, while opportunities and threats are external attributes present in the outside environment. Strengths and opportunities help the program achieve its goal, and, conversely, weaknesses and threats hinder the program’s success. Like conjoint analysis, SWOT analysis was originally developed for business management but has been adapted by the health sciences to evaluate health control programs [14]. This tool has previously been used by the veterinary sciences, such as the evaluation of alternative control strategies for ASF [15]. In the Philippine context, SWOT analysis can support the identification of important weaknesses and threats to controlling ASF.

The objective of this study was to gain insights into the risk factors associated with ASF in the Philippines by utilizing SWOT and conjoint analysis. Additionally, we sought to assess the ASF clinical presentation and industry control measures in both backyard farms and commercial farms, relying on the expertise of professionals in the swine industry. While acknowledging the inherent limitations of expert opinion-based research, these findings provide valuable insights that could potentially contribute to the enhancement of surveillance plans and policy development aimed at addressing ASF risks. Ultimately, this work endeavors to contribute to ongoing efforts in improving ASF control in the Philippines, with potential implications for safeguarding the swine industry and public health.

## 2. Materials and Methods

### 2.1. General Approach

To support the identification of ASF risk factors, a group of veterinary experts specializing in the Philippine swine industry and with field experience with ASF was selected through a committee evaluation process. A workshop using a modified world café approach was organized with the aim of facilitating knowledge sharing and education among these participants, as well as identifying research gaps in ASF control. Conjoint analysis and SWOT analysis were used to gather expert insights and elicit their opinion. These methodologies allowed the experts to engage in brainstorming sessions and collaboratively identify the current perspectives on the spread, diagnosis, and control of ASF in the Philippines, accounting for its endemic status for over four years.

### 2.2. Expert Selection

To select participants for the workshop, a committee was assembled by the Philippine College of Swine Practitioners (PCSP), the Department of Agriculture Bureau of Animal Industry (DA-BAI), and the Agricultural Training Institute-International Training Center on Pig Husbandry (ATI-ITCPH). The committee utilized specific criteria to evaluate the suitability of applicants. To be eligible for selection, applicants were required to have 1–2 years of experience in the Philippine swine industry, but preference was given to those having 5 to 10 years of experience. Additionally, applicants were required to either currently hold or have the potential to become decision makers, such as regulatory authorities in their respective areas. The committee also specifically aimed to have at least five participants from each of the Department of Agriculture Regional Field Offices, the DA-BAI Policy Division, local governmental units, and the private sector. Finally, the committee gave special consideration to ensuring balanced representation from each region. After an initial open call for applicants, approximately 100 applicants were considered in total, and 25 were selected for the final workshop. This process took place between February and March 2023, and the workshop was held in May 2023.

A group of 25 selected veterinarians from different spatial locations across the Philippines (Figure 1) attended the in-person workshop organized in Batangas in May 2023. The National Capital Region (NCR) had 7 participants, Region IV-A had 6 participants, and the remaining regions were evenly distributed. Ten participants were from the Philippine veterinary services, and 15 were from the private Philippine swine industry. The participants had a median of 20 years of experience in swine veterinary practice or service within the country. All participants indicated that they had some degree of field experience with ASF and that they have seen the disease.

### 2.3. Conjoint Analysis

Conjoint analysis was used to identify and prioritize factors associated with the introduction and transmission of ASF in the Philippines. A list of nine risk factors important to African Swine Fever introduction and spread were identified based on existing research [16,17,18]. Each factor was dichotomized into a high- or low-risk response (Table 1).

A ranking-based survey was developed to estimate the individual influence of each factor on the risk of an ASF outbreak. Because it would be virtually impossible for participants to rank all combinations of these factors, a factorial design was generated to select a subset of farm scenarios. The scenario selection process was performed using the orthogonal design and conjoint plan methodology available in IBM SPSS Statistics v28.0.0.0. The final conjoint experimental design incorporated 13 different scenarios, with best- and worst-case scenarios included as control groups (Figure 2).

The participants were Individually surveyed to assess and prioritize the risk and likelihood of ASF outbreaks in 13 hypothetical farm scenarios (Figure 2) using the Qualtrics platform. The participants were tasked with ranking these farm scenarios based on two criteria: (1) the level of risk associated with each farm ranging from 1 (representing the lowest risk of an ASF outbreak) to 13 (the highest risk); and (2) the likelihood of an ASF outbreak occurring using a binary response format (Yes or No).

### 2.4. Quantitative ASF Information Collection through World Café Discussion

A modified Delphi approach was implemented in a world café format discussion to foster open and creative dialogue among the participants and validate received responses. The participants were divided into four groups to ensure comprehensive discussions. After a designated time, participants were asked to switch tables and engage in discussions on the additional sets of questions. Each round involved a randomly ordered sharing of ideas, and a designated reporter summarized the group’s consensus while seeking agreement or dissent from other groups. A modified Delphi method was employed to gather further information on key parameters and to validate the answers received from previous groups.

The modified world café sessions included quantitative questions that focused on three main areas, namely, (a) the routes through which ASF enters farms; (b) clinical manifestations and necropsy findings associated with ASF in the field; and (c) strategies and policies for ASF control in backyard and commercial farming. For the details of quantitative questions, please refer to the Appendix A provided.

### 2.5. SWOT Analysis

SWOT analysis was used to gather insights and perspectives on the current status, diagnosis, and control policies of ASF in the Philippines. Participants were asked to identify strengths, weaknesses, opportunities, and threats associated with these factors as part of the previously described modified world café discussion. The SWOT table was summarized by the authors and presented to the participants during the workshop for their agreement and further discussion. Throughout the process, all discussions and outcomes from the modified world café sessions and SWOT analysis were documented for reference and subsequent analysis.

### 2.6. Statistical Analyses

#### 2.6.1. Logistic and Ordinal Regression Models

Participants’ responses from the conjoint analysis survey (Figure 2) were analyzed using ordinal and binary logistic regressions. The response variables were whether participants believed a scenario farm ID (A to M) would lead to an ASF outbreak or not (Yes/No data for logistic regression) and the participants’ individual ranking of perceived risk for each scenario (1–13 rankings data for ordinal regression). We created datasets for the analysis by matching each response with the corresponding scenario farm ID and its risk factor combination (Table 1). The dichotomous risk factors associated with each scenario were used as predictor variables in the regression models to predict the participant response.

Binary logistic regression and ordinal logistic regression models were performed in RStudio version 4.2.2 using the MASS package [19]. As an output of the regression analysis, regression coefficients were obtained for each risk factor, which provides an estimate of the relative weight each factor has on the responses. These coefficients were used to understand the significance and impact of the different risk factors under investigation on the probability of an ASF introduction on Philippine farms.

The Hosmer and Lemeshow goodness of fit (GOF) was performed using the ResourceSelection package [20]. The GOF test evaluates if a logistic regression model fits the observed data, determining if it could accurately represent the relationship between independent variables (risk factors) and the dependent variable (ASF occurrence, in this case).

#### 2.6.2. Statistical Analyses World Café Discussion and SWOT Analysis

Discussions from the SWOT analysis were summarized using a SWOT (Strengths, Weaknesses, Opportunities, Threats) analysis table. We compiled the discussions from four groups and allowed sufficient time for participants to reconsider and reflect on their opinions. Our focus was on documenting the information that achieved consensus and was supported by multiple opinions, aiming to maximize the consensus in the SWOT analysis and seek detailed explanations for each characteristic. While this approach does not involve calculating the SWOT analysis numerically, it allowed us to capture the majority of opinions effectively.

For the quantitative questions in the world café sessions, each table was required to reach a consensus within their randomized group. Subsequently, median values and normalized median scores were estimated for questionnaire results in this session.

## 3. Results

### 3.1. Conjoint Analysis

The logistic regression analysis identified swill or contaminated feed as the most significant concern (*p* < 0.001) across all 13 scenarios (Table 2). Following that, the absence of a protocol for disinfection (*p* = 0.0017) and personnel lacking trusted biosecurity measures (*p* = 0.0019) were also identified as significant risks. In the ordinal logistic regression model, all risk factors showed significant associations with increasing the probability of a scenario being higher ranked (i.e., higher perceived risk of ASF introduction by the participants; Table 2). The common odds ratios provide information on the magnitude of these associations, indicating how much the odds of moving up to a higher category change with each unit increase in the predictor variable. The highest common odds ratio, with a value of 19.84 (95% CI 12.09–33.22), was observed for swill-fed or contaminated feed, followed by personnel lacking trusted biosecurity measures (11.11, 95% CI 6.88–18.26) and absence of a protocol for disinfection (7.54, 95% CI 4.73–12.21). These results from the ordinal logistic regression were consistent with the binary logistical regression, with the top three risk factors being swill-fed/contaminated feed, personnel, and the absence of a protocol for disinfection.

The Hosmer and Lemeshow GOF test resulted in a *p*-value of 0.8487, suggesting a good model fit for the data.

### 3.2. World Café Discussion

The activity of estimating infection routes by participants revealed distinct patterns for commercial farms and backyard farms, as indicated by the summary statistics from the four discussion groups (Table 3). For commercial farms, participants believed contaminated vehicles and people were the primary risk factors (49.8% of the hypothetical cases). In contrast, participants indicated that backyard farms face a more diverse range of risks. These include importing new pigs, environmental contamination, vehicles, and concerns regarding feed quality. The participants considered wild boars to pose the lowest risk among all the factors considered in both types of farms.

In terms of clinical presentation, the most prominent clinical signs typically observed in pigs with ASF include a drop in feed intake, huddled pigs, and reluctance to stand up (Table 4). Participants reported that these clinical signs should typically be detected by swine producers. However, they indicated that hematemesis (vomiting blood) and constipation followed by diarrhea have a lower probability of being detected by producers.

For necropsy findings, participants reported that splenomegaly (100%), kidney hemorrhages (80%), and lymphadenomegaly (65%) are among the most prominent observations. Conversely, they felt that the likelihood of producers noticing these signs is low (25%). Participants reported that they less frequently observe pneumonia, hydropericardium, hydrothorax, and shock lung/ARDS in the field, with a median percentage below 15%.

Participants also provided their estimates of the timeline of clinical signs and necropsy finding appearance during ASF infection (Figure 3). According to respondents, affected pigs initially show reduced activity and decreased feed consumption (Day 6). As ASF progresses, they develop fever (Day 7), huddle together (Day 7), and exhibit erythema, or reddening of the skin (Day 8.5). In pigs that survive to later stages, bleeding from the nose, mouth, or rectum (Day 14) and cyanosis (Day 14.5), a bluish discoloration of the skin and mucous membranes may occur.

Notably, participants reported that ASF clinical signs may be observed as early as Day 6. For necropsy findings, the earliest significant findings are usually observed approximately 14 days after ASF infection (Figure 3, coral orange bars).

Participants also reported on strategies and policies implemented for ASF control by producers. According to estimates from the participants, both commercial and backyard farmers frequently choose to sell their pigs (95%) once ASF is detected to prevent economic losses from total depopulation (Figure 4), which is mandated by Philippine government policies. Notably, the experts estimated that total depopulation is higher in backyard farms (80%) than in commercial farms (50%), but reporting rates of ASF cases are higher among backyard farmers (20%) compared to commercial farms (10%).

### 3.3. SWOT Analysis

The SWOT analysis (Table 5) of ASF control in the Philippines identified strengths such as the National Control Policy and collaboration between government levels. Weaknesses included a lack of compensation and trust, the absence of a traceability system, and resource constraints. Opportunities included using social media and responsible technology, while threats encompassed movement risks and limited veterinarians.

## 4. Discussion

Despite ASF spreading within the country since 2019, field data describing the clinical presentation of the disease and the factors affecting disease spread and control in the Philippines are limited, or where present, difficult to access. This may be explained, at least in part, by the challenges associated with collecting and reporting accurate data, such as knowledge of and access to affected farms, the potentially sensitive nature of collected data, and the availability of necessary resources and personnel for data collection. To better understand ASF in the Philippines, a group of practitioners and government officers with field experience with the disease was assembled, and a mixture of qualitative and quantitative methods were used to gather their collective opinion on important features of ASF outbreaks and spread. This paper presents, for the first time, information on the factors affecting the spread, diagnosis, and control of ASF in the Philippines.

Results of the conjoint analysis suggested that swill or contaminated feed would be the most significant factor influencing ASF spread, with a 6.51-fold increase in the odds of ASF occurrence compared to its absence (Table 2). This finding is consistent with previous research in Asia [21,22], indicating that despite government bans, the risk of swill feeding remains significant. However, when examining Table 3 and discussing the risk of ASF transmission in the Philippines, the perceived importance of swill feeding was not as high. The follow-up discussion revealed that the discrepancy was because of a reduction in the frequency of swill-feeding in the Philippines since the beginning of the epidemic. Thus, even though swill feeding is considered the most important factor affecting ASF spread in the country because most farms have reduced the use of swill for feeding, in the event of an outbreak, it is likely that other routes may be responsible for disease spread.

The second highest concern in the swine industry relates to human behavior, particularly the movement of personnel (such as veterinarians, technicians, and workers) between farms without proper biosecurity measures. This factor was substantiated by the data presented in Table 3, and the concern is the entry of contaminated individuals onto the farm, surpassing other risks. Additionally, during the world café discussion and SWOT analysis, the presence of hog traders, also known as middle men, intermediaries or “viajero” locally, was emphasized in the supply chain. Participants agreed that backyard farmers or small commercial producers, who lack direct access to customers, heavily rely on intermediaries or hog traders to facilitate pig sales. This reliance increases the risk of disease transmission, especially if proper biosecurity measures are not followed between farms.

Previous research has shown that the problem of wild boar and its relationship with ASF is underestimated in Asia, and the Philippines has been categorized as a medium-risk level [23]. Based on our conjoint results and world café findings (Table 3), the risk associated with wild boar or feral pigs is not a major concern, at least indicated by the consensus from the world café discussion. In the NCR region, the chance of domestic pigs encountering wild boar is low. However, in regions with higher wild pig populations like Ilocos (Region I), Cagayan Valley (Region II), and Cordillera Administrative Region, an increased likelihood of wild boar contact may exist [24]. Due to the lack of effective control measures for freely roaming cats and dogs [25,26], stray cats and dogs could act as potential mechanical vector animals for ASFV in the Philippines. The accessibility of backyard farms and the shift towards open fencing make it easy for stray animals to enter farms, access swine feeds and carcasses, and move between different farms within the cities. To prevent access from scavengers such as cats and dogs, burying swine carcasses with lime is suggested [27]. These roaming animals may potentially contribute to the spread of ASF, but as of yet, no studies have documented a major role of these animals in ASF spread. Improved biosecurity measures on farms will help keep out unwanted pets and wildlife from swine areas.

The participants also believed that swine producers could recognize most clinical signs associated with ASF in the Philippines. These findings are consistent with previous pathology research [28]. The necropsy findings provide additional confirmation, with splenomegaly, kidney hemorrhages, and lymph node enlargement being the most commonly observed indicators of the disease. However, the participants felt that pneumonia, hydropericardium, and acute respiratory distress syndrome are less frequently identified. This may be because they are associated with later stages of ASF that are not commonly seen. The timing of culling and necropsy procedures, which primarily occur during the early phases of ASF infection, may minimize the frequency of these signs being observed. This early intervention allows for the rapid detection and depopulation of affected swine, reducing the likelihood of severe necropsy findings during the examination by veterinarians. Drop in feed consumption seems to be an important early sign of infection and may be observed by producers as early as 6 days post infection (Figure 3, Table 4). While not specifically explored in the workshop activities, the likelihood of producers noticing specific clinical signs or necropsy findings may be due to differences in producers’ individual competency and training.

In terms of control strategies, the participants felt that commercial farms typically lean towards partial depopulation, whereas backyard farms have a higher rate of total depopulation. This may reflect an advantage of commercial farms, as typically they possess better resources and capabilities to manage and mitigate losses during ASF outbreaks. They have established protocols and resources in place to handle the disposal of infected pigs and implement necessary biosecurity measures. In contrast, backyard farmers often rely on government support, particularly for compensation, during ASF outbreaks. Additionally, the smaller pig populations in backyard farms make it easier for them to conduct total culling compared to partial depopulation.

The experts’ estimate of a low reporting rate for both commercial and backyard farms (Figure 4) highlights the need to improve ASF reporting for all farm types, such as by providing incentives for timely reporting. As observed during the workshop discussions, this untimely reporting is mainly due to a lack of trust among farmers regarding the effectiveness of government assistance. For example, farmers in both commercial and backyard settings bear the financial burden of burying dead pigs and processing affected animals, which can decrease their willingness to accurately report ASF cases. Concerns about the potential spread of ASF may lead farmers to resist the entry of government officers or veterinarians. These disparities in resources and perceptions highlight the challenges farmers face in managing ASF outbreaks effectively. Weaknesses identified in the SWOT analysis, such as the lack of compensation and trust, further contribute to farmers’ perception of inadequate support. To address these issues, the government should develop an early reporting system and prioritize trust-building initiatives between swine farmers and government partnerships.

This research is limited by the potential introduction of sampling bias through the expert selection process, which may inadvertently favor certain perspectives and exclude insights from other stakeholders in the swine industry, including that of producers, for example. However, because much of the experience gained by veterinarians while fighting the disease in the field is not captured in official records (due to underreporting or limitations in the quality of data collected) or scientific publications, elicitation of their opinion is often the most valuable alternative to collect, organize, and share their collective experience with the disease in the field. Although efforts were made to ensure diversity following selection criteria, the final selection of 25 veterinarians, while reflecting a significant portion of industry opinion, may not fully represent the entire range of perspectives and experiences related to ASF in the Philippines. As a result, the findings and conclusions of this research may not encompass the complete spectrum of knowledge and viewpoints on ASF control. Despite these limitations and potential sources of bias, the publication here offers an opportunity to access and evaluate, in an organized way, the consensus that may have been reached in the community of veterinary practitioners in the Philippines working on the control of the disease in the field. For that reason, these results provide valuable input to supplement the assessment of “hard” outbreak data that may be collected in the field, which, by nature, may also be biased and limited.

In summary, the study findings and expert estimates suggest that human behavior might be the most important factor affecting the spread of ASF. The elicitation of the opinion of practitioners with field experience fighting ASF in the Philippines suggested that the most important factors for ASF introduction were swill feeding, movement of personnel without proper biosecurity, and the absence of disinfection protocols, while wild boar are only a concern in regions with higher feral pig populations. This consensus suggests that any effort made to avoid the entrance of vehicles or individuals into susceptible farms may be highly impactful in preventing disease spread in the Philippines. Overall, these findings contribute to a better understanding of ASF spread, diagnosis, and control. They may support government policy development in the Philippines and contribute to enhancing preparedness in ASF-free areas worldwide.

## Figures and Tables

**Figure 1 pathogens-12-01068-f001:**
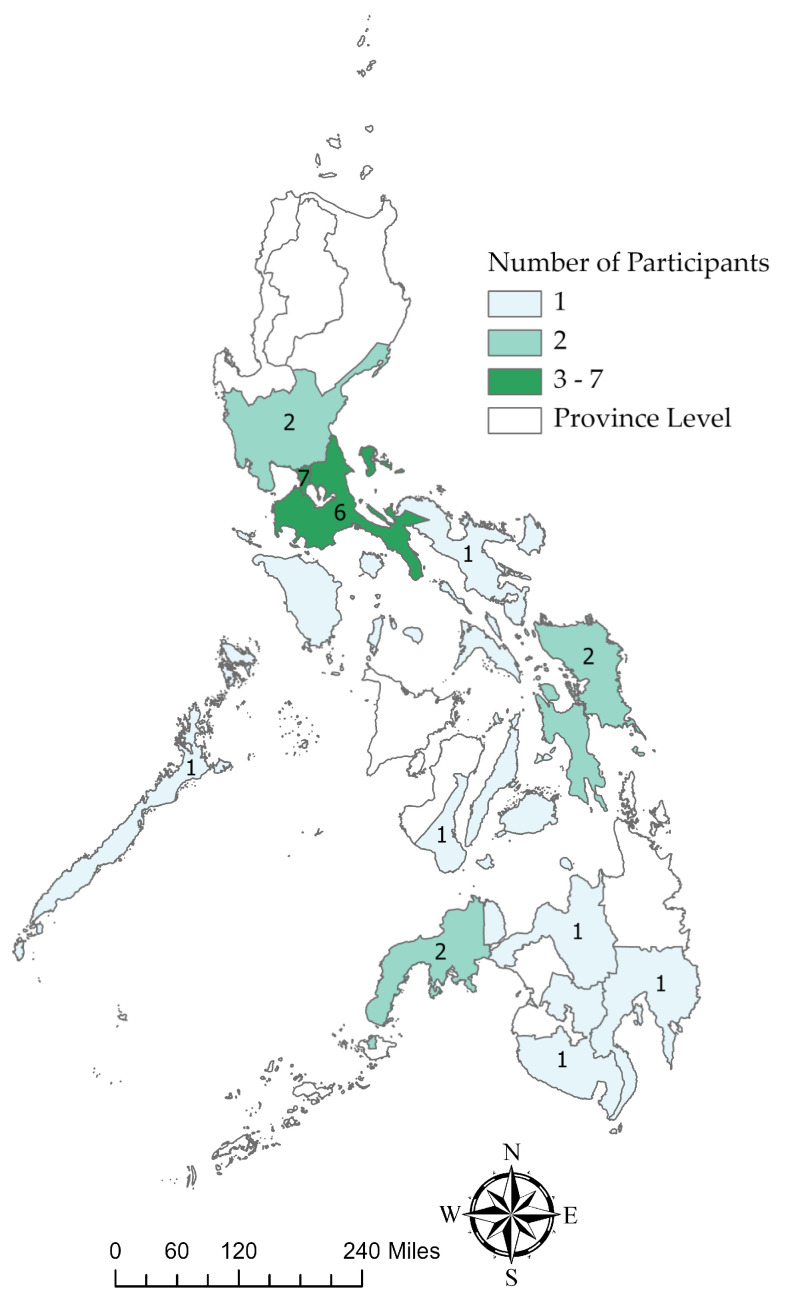
Geographic distribution of workshop participants in the Philippines.

**Figure 2 pathogens-12-01068-f002:**
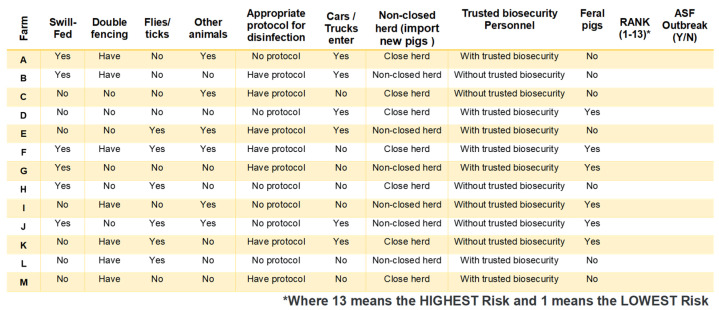
Scenarios for ranking-based conjoint analysis of ASF introduction risk.

**Figure 3 pathogens-12-01068-f003:**
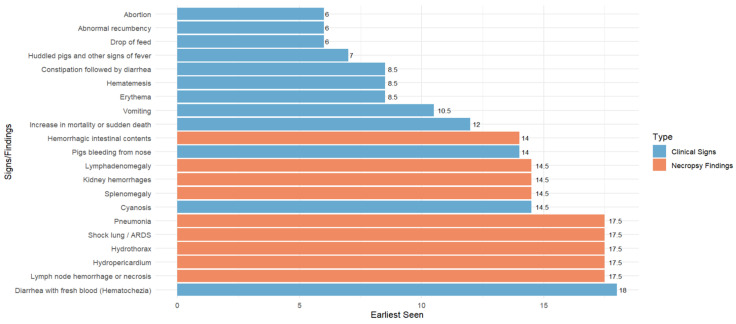
Consensus reached from the elicitation of expert opinion on the earliest timeline of clinical signs and necropsy findings during ASF infection.

**Figure 4 pathogens-12-01068-f004:**
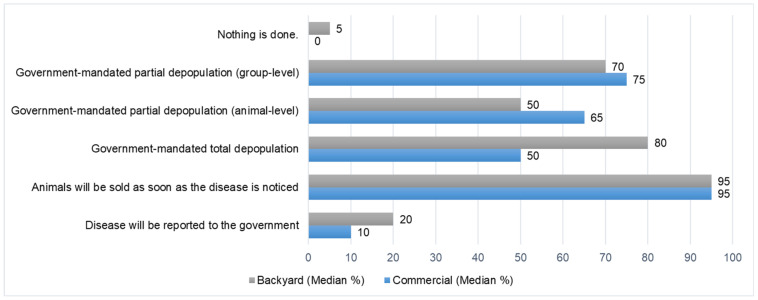
Consensus reached from the elicitation of expert opinion on the estimated control actions by commercial and backyard producers following an on-farm ASF detection.

**Table 1 pathogens-12-01068-t001:** Risk factors and their levels associated with ASF introduction into farms for conjoint analysis.

No.	Risk Factors	High-/Low-Risk Response
1	Swill-fed or potential contamination of feed ingredients	Yes/No
2	Lack of double fencing	No double fencing/Have double fencing
3	Presence of flies and ticks	Yes/No
4	Presence of small and domestic mammals (e.g., rats, dogs, cats, or other farm animals)	Yes/No
5	Absence of protocols for changing clothes, separate entries and exits, disinfection of objects restrictions on food introduction, and external individuals accessing the farm	No appropriate protocols/Have appropriate protocols
6	Allowance for cars and trucks to enter premises	Cars and trucks can enter premises/Cars and trucks cannot enter premises
7	Non-closed herd with recent introduction of new animals (requiring importation of pigs) without a quarantine station within 1 km from premises or sharing of personnel	Non-closed herd/Closed herd
8	Movement of personnel (including vets, inseminators, and technicians) between this farm and other farms without trusted biosecurity measures	Personnel without trusted biosecurity measures/Personnel with trusted biosecurity measures
9	Area with presence of feral pigs	Yes/No

**Table 2 pathogens-12-01068-t002:** Conjoint analysis results: logistic regression and ordinal logistic regression models.

	Logistic Regression Model						Ordinal Logistic Regression					
Description	Coefficient	CI (95%)	Odds Ratio	SE	*p*-Value	Sig.	Coefficient	SE	Common Odds Ratio	CI (95%)	*p*-Value	Sig.
Swill-fed/Contaminated feed	1.8740	(1.28, 2.54)	6.5143	0.3167	<0.001	***	2.988	0.258	19.8459	(12.09, 33.22)	3.92 × 10^−31^	***
Absence of double fencing	0.4526	(−0.13, 1.09)	1.5723	0.3067	0.14001		1.611	0.235	5.0078	(3.18, 7.99)	6.57 × 10^−12^	***
Presence of flies and ticks	0.7355	(0.15, 1.38)	2.0865	0.3080	0.01696	*	1.936	0.237	6.93	(4.38, 11.12)	3.56 × 10^−16^	***
Presence of other animals	0.8089	(0.26, 1.38)	2.2454	0.2866	0.00476	**	1.558	0.231	4.749	(3.04, 7.53)	1.57 × 10^−11^	***
Absence of a protocol for disinfection	0.9682	(0.38, 1.61)	2.6332	0.3092	0.00174	**	2.02	0.242	7.538	(4.73, 12.21)	6.61 × 10^−17^	***
Cars/trucks enter	0.4669	(−0.07, 1.02)	1.595	0.2779	0.09293		1.401	0.232	4.059	(2.59, 6.44)	1.45 × 10^−9^	***
Non-close herd	0.3551	(−0.20, 0.92)	1.4263	0.2829	0.20937		1.693	0.231	5.4357	(3.48, 8.61)	2.45 × 10^−13^	***
Personnel without trusted biosecurity	0.9593	(0.37, 1.60)	2.6099	0.3094	0.00193	**	2.408	0.249	11.11	(6.88, 18.26)	3.51 × 10^−22^	***
Area with feral pigs	0.4925	(−0.06, 1.05)	1.6364	0.2825	0.08126		1.463	0.233	4.3189	(2.75, 6.87)	3.40 × 10^−10^	***

* *p* < 0.05; ** *p* < 0.01; *** *p* < 0.001.

**Table 3 pathogens-12-01068-t003:** Consensus reached from the elicitation of expert opinion on the probability of ASF introduction route for commercial and backyard farms.

Infection Route	Commercial (Normalized Median, %)	Backyard (Normalized Median, %)
Introduction of sick pigs	9.9	15.8
Environmental contamination (water courses, rice fields next to the farm, contact with backyard farms, etc.)	17.7	13
Contaminated vehicles entering the farm	29.6	16.4
Contaminated people entering the farm	20.2	22.6
Feed	7.9	13.6
Wild boars	0	5.1
Rodents, flies, and other potential vectors	14.8	13.6
Other	0	0
Total	100	100

**Table 4 pathogens-12-01068-t004:** Consensus reached from the elicitation of expert opinion on the frequency and the likelihood seeing clinical signs and necropsy findings.

Clinical Sign	Median Frequency (%)	Median Likelihood of Producer Seeing (%)
Drop in feed consumption	99.5	100
Huddled pigs and other signs of fever	85	100
Reluctances to stand up and move (abnormal recumbence)	85	100
Reddish in the skin (Erythema)	65	100
Abortion	65	100
Increase in mortality or sudden death	64.5	100
Diarrhea with fresh blood (hematochezia)	35	100
Cyanosis (blue areas) of ears and limbs	30	100
Pigs bleeding from nose	28	100
Vomiting with blood (hematemesis)	12.5	75
Constipation followed by diarrhea	10	25
Vomiting	5	100
Necropsy Findings	Median Frequency (%)	Median Likelihood of Producer Seeing (%)
Splenomegaly	100	25
Kidney hemorrhages	80	25
Lymphadenomegaly	65	25
Lymph node hemorrhage or necrosis	50	25
Hemorrhagic intestinal contents	50	66.7
Hydropericardium	11	25
Hydrothorax	11	25
Shock lung/acute respiratory distress syndrome	10	25
Pneumonia	10	25

**Table 5 pathogens-12-01068-t005:** Summary of SWOT results from world café discussion.

Strengths	Weaknesses
1. National Control Policy and ASF task force. 2. Collaboration between different levels of government (central level and local governmental units). 3. Evidence-based approach. 4. Philippines Statistics Authority has a regular report of inventory of swine and farmers, including type of production.	1. Farmers face difficulties due to the absence of adequate compensation and a lack of trust in the government’s support. This leads to resistance to reporting ASF cases due to the absence of incentives and negative public opinion. 2. The value chain lacks an established traceability system, making it difficult to track and map the routes of hog traders (viajeros), which hinders effective control measures. 3. Lack of resources. There are significant resource constraints in terms of manpower and testing capacity. Limited manpower affects the implementation of control measures, and although the testing capacity has improved, the efficiency of diagnostic PCR testing (RADDL) results in delayed reporting, which hampers timely control efforts.
**Opportunities**	**Threats**
1. Social media such as Facebook fan page or TikTok for dissemination of ASF prevention information. 2. Rapid adoption of responsible technology in diagnostics. 3. Environmental compliance and regulatory practices for related industries. 4. Recent vaccine trials. 5. Improvement in the execution of biosecurity measures and culture.	1. Risk factors such as human/trade movement and vectors in the Philippines. 2. Dwindling number of new swine veterinarians. 3. Issues with slaughterhouse compliance, tampering with documents, and border control corruption. 4. Time to detection of ASF.

## Data Availability

The data, which are the property of the Philippines government and have been shared with us to conduct research here in support of training activities in the country, can be obtained on request from the corresponding author. For inquiries, kindly direct requests to the Philippine Bureau of Animal Industry and the International Training Center on Pig Husbandry of the Philippines.

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
