# Peer review of "Factors Affecting the Spread, Diagnosis, and Control of African Swine Fever in the Philippines"

_pathogens, 2023, doi:10.3390/pathogens12081068_

Round 1
Reviewer 1 Report
The work is written in a logical way, in the sense that the objectives of the study set out in the introduction have been achieved and have given results. Even if some points of the materials and methods can be questionable (such as the number of subjects interviewed compared to the possible ones present in the area; number of companies/farms of which we have information compared to those present in the area). However the work highlights the most factors probable determining risk of infection in the Philippines.
it is understood that the important significance of this work is that it is in any case a starting point to better control the spread of the infection through the implementation of prevention practices. This it does by highlighting the points of lack of prevention and bringing to light what is missing and the incorrect actions that facilitate the diffusion. It is a sort of snapshot of the socio-economic and social health situation of pig farms in the Philippines.
Author Response
Response to Reviewer 1 Comments
Thank you, Reviewer 1. We extend our gratitude for your valuable suggestions, which have been instrumental in guiding us through the major revision process. Your insights have been thoughtfully integrated into the revised manuscript, and we strongly believe that your contributions have markedly improved the overall quality of our work. In response to your feedback, we have diligently revised the manuscript and further elaborated on the limitations of our study. This has allowed us to approach our revision with a sense of humility and transparently acknowledge the constraints of our research.
Please note that our revisions are highlighted in red in the Word document.
Reviewer 2 Report
The manuscript is written in an understandable and logical way. The aim of the text covering ASF risk and SWOT analysis and efficiency related to ASF activities were presented correctly. The work is based on properly selected literature, although the references included should be supplemented and expanded. The tables should be unified to the same form, while graphics no. 1 should be enlarged.
Author Response
Response to Reviewer 2 Comments
Thank you, Reviewer 2. Your feedback has significantly improved our manuscript's coherence and logical flow. We've reviewed and selected appropriate references, standardized our table format, and coordinated with the editor for consistent formatting. Thank you for your valuable contribution.
Please note that our revisions are highlighted in red in the Word document.
Reviewer 3 Report
Please see attached document

The English is good, only minor errors detected.
Author Response
Response to Reviewer 3 Comments
Thank you, Reviewer 3, for your constructive and detailed suggestions, which have played a crucial role in guiding us through the major revision process. We have meticulously incorporated your insightful recommendations into the revised manuscript, and we firmly believe that your contributions have significantly enhanced the overall quality of our work. Once again, we express our gratitude for your time, expertise, and dedication to ensuring the rigor and relevance of our research. We anticipate submitting the revised manuscript and sincerely hope that it aligns with the high standards you have helped us strive for.
Please note that our revision is highlighted in red in the Word document.
Introduction
Line 47-48: It is not entirely clear to me what is meant by a 41.7% decline in pig inventory, experienced by commercial farms. Does this refer to the entire population of pigs in commercial holdings? I suggest rephrasing to …”In 2021, there was a 41.7% reduction of the total population on commercial pig farms, with a subsequent inflation of pork prices”. Likewise, in the subsequent sentence I suggest replacing “inventory” by “population” (or “registered population” if that is what you mean). Line 50-53: As you describe a national pig population mainly kept on (non-commercial?) small holdings, a reference to how this sector was hit by ASF seem appropriate. As I understand it now, the >40% decline of the population took place on (large) holdings keeping less than half of the pigs in the country.
Revisions made in lines 47-52.
Line 57: please add “and” before “…ensure…”
Revisions made in lines 55.
Line 61: Please add “been” to read …”has not been formally recorded.”
Revisions made in lines 60.
Line 68-72: The second half of this sentence gives a confusing description of statistical methods. I presume it is quoted from reference no 11 but this source does not seem to be publicly available (login required to access via provided link). I suggest removing the second half of the sentence, or shorten and rephrasing it to what you actually intend to convey.
Revisions made in lines 67-74.
Line 88-91: The objectives were very ambitious; did you really expect to be able to assess the effectiveness of control measures based only on expert opinion? I suggest modifying to make the objectives more modest and realistic.
Revisions made in lines 89-98.
Materials and Methods
Line 108: You refer to “applicants”, this sounds like a call went out and experts could apply, is that correct? Or were they selected from lists, personal knowledge of the committee members? Please clarify the initial phase of the selection process.
Revisions made in lines 122-124.
Line 137: I suggest adding this reference, which is highly relevant for the situation and more recent than no 18: https://www.fao.org/publications/card/en/c/CB9187EN
Agree with this point. This new reference has already been modified as Reference 18.
Revisions made in lines 140.
Line 152-153: Why use a binary response for the likelihood of an ASF outbreak? Wouldn’t a high/low/medium likelihood be easier to provide (and more reliable)? Please justify.
Although high/low/medium classifications may initially appear intuitive, they introduce complexities in data analysis and interpretation. Assigning appropriate “thresholds” for these categories can be challenging, resulting in varying results and making comparisons between studies more difficult.
By adopting a binary response (logistic regression), we ensure consistency and enable straightforward statistical analysis, enhancing the reliability and objectivity of the study's outcomes.
We have conducted two separate analyses, one for the dichotomous response and another for the ranks. Considering the nature of this response, categorizing it into three distinct categories might introduce some redundancy in terms of analysis. This is due to its intermediary position between the two responses being analyzed, which are categorization into two classes and the use of ranks. Furthermore, the similarity of results obtained from both models (ranks and dichotomization) suggests that grouping into three categories could potentially yield similar outcomes. Therefore, it appears that adopting three categories might introduce an element of redundancy.
Line 178-193: It is difficult to understand what was done in the statistical analyses, how did you include the risk factors in regression models? Was there a weighted score for each factor or how did you include them in a statistical model? As you have described the exercises, scenarios were ranked but not individual risk factors, and these scenarios were assessed as to whether they would result in an ASF outbreak or not. It is not sufficient to refer to a text on conjoint analysis, the statistics must be described so as to be comprehensible (and possible to assess). Although this is common in conjoint analysis, how do you justify the use of regression models based on ranked numbers (instead of non-parametric methods)? Did you treat the ranks as ordinal data? How did you quantify the SWOT data so that descriptive statistics could be calculated? A more thorough explanation is warranted. I find the statistics section difficult to grasp and not convincing as regards scientific rigour. I confess that I am not experienced in conjoint analysis but unless you provide more explanation of the methods and justification for the statistics I am not convinced. You are essentially assessing the experts’ opinion based on their own estimates… The use of these methods in references 12 and 13 is different from what you have done, one assessed the figures from ranking against respondent characteristics and the other used a similar approach as yours but then validated the models against real data. You have not done the latter part but use your models as if they provide real field data, which makes the entire paper doubtful.
We rewrite the section to enhance clarity.
Revisions made in lines 185-215.
Results
Line 217-218: How can you state that these routes accounted for a certain proportion of the cases? Wasn’t this a discussion among experts on what they thought were the main causes/routes of infection? There were no real “cases” and no way to “account for” these.
Revisions made in lines 238-241.
Table 3: how were these numbers produced, are they based on the estimates of the different groups? If so, did you use outputs from a group consensus or individual estimates from the group members?
We have revised the title of Table 3 to “Consensus reached from the elicitation of expert opinion on the probability of ASF introduction route for commercial and backyard farms.”
Revisions made in lines 246-247.
Lines 226-227: I suggest rephrasing to stating that these signs were regarded as less likely to occur in ASF cases, instead of it being less likely to be detected, as you were in fact asking for the signs expected and not the probability of the farmer being able to detect them. Or did the question infer the ability of the farmer to observe the signs? This applies also to lines
Revisions made in lines 248-252.
Lines 228-232: Are the statements about these symptoms being less likely to be observed by the farmer due to the fact that they don’t appear as specific obvious clinical signs? Or do you mean that the producers lack the ability to identify them? I find it very surprising that the producer would observe either kidney haemorrhage or splenomegaly, do they perform field necropsy of dead pigs themselves?
Revisions made in lines 253-258.
Line 250: what do you mean by “ratio” here? A percentage is not a ratio, it’s a proportion but what do the figures mean? It seems that these are estimates by the study participants, but it is not clear if they refer to voluntary depopulation or actions enforced by law. It must be clearly expressed that these are estimates/beliefs from the participants, you cannot state that “the ratio is” and “there is a tendency” unless you have solid data on this and refer to those. In this case, you must write that “according to the estimates by the participants….” or something similar.
Revisions made in lines 278-280.
Discussion
Line 266-267: Field data on the clinical presentation and factors that affect spread and control may not be overwhelming, but a number of papers have described these, since the introduction and spread of ASF in Asia. Please modify the statement that field data are scarce. It is still of interest to know this for each country, as it is needed for awareness campaigns, understanding the local impact and addressing local challenges and you may write that more field data are needed (although they may not really be regarded as “scarce”).
Line 268: replace “at list” by “at least”
Revisions made in lines 294-304.
Line 313: Did the study really reveal this about the producers, wasn’t it the experts who thought so?
Revisions made in lines 343-358.
Line 332-333: The explanation may be true, but you do not have a “resulting ratio of depopulation” here, you have expert opinion on how likely/frequently (still not clear to me but not a ratio) it is to occur in different types of farms.
Revisions made in lines 365-367.
Line 334: “the finding of a low reporting rate” is another example of not treating the data correctly, you do not have hard data on reporting rates, you have estimates/expert opinion on these.
Revisions made in lines 368.
Line 347-353: I suggest also mentioning the absence of farmers’ perspectives, you have elicited the opinion of veterinarians about farmer behaviour and speculate about the reason for this behaviour but unless you talk to farmers (and/or collect data on their behaviour) these remain opinions and speculations. The Discussion is generally better phrased than the Results section, but it is very important to not describe expert opinion as factual data. In this case, the experts have discussed together, and it is not unfathomable that they would have convinced each other based on the experiences of some participants so that they all agreed on some things that not everyone had experienced. Some reflections on the choice of statistical methods and how representative your “data” may be should also be added.
Revisions made in lines 382-399.
Conclusions
The conclusions are too strongly phrased and I recommend a more modest way of putting forward the still very interesting results. Remember that you have elicited expert opinions and not hard data, and the results stem from only one professional category. Although well selected, the participants cannot reflect the entire spectrum of experiences of ASF in the Philippines and this must be borne in mind and shown in the way data are presented and discussed.
Revisions made in lines 400-410.
Round 2
Reviewer 3 Report
Thank you for addressing my concerns. I am content with the manuscript in its current version.